# Survey of the Adequacy of Brazilian Children and Adolescents to the 24-Hour Movement Guidelines before and during the COVID-19 Pandemic

**DOI:** 10.3390/ijerph20095737

**Published:** 2023-05-08

**Authors:** Natália Molleri, Saint Clair Gomes Junior, Daniele Marano, Andrea Zin

**Affiliations:** National Institute of Women’s, Child’s and Adolescent’s Health Fernandes Figueira, Rio de Janeiro 22250-020, Brazil; natalia.molleri@iff.fiocruz.br (N.M.); saintclair.junior@iff.fiocruz.br (S.C.G.J.); danielemarano@yahoo.com.br (D.M.)

**Keywords:** guideline adherence, exercise, screen time, sleep, infant, preschool, child, adolescent

## Abstract

The 24-Hour Movement Guidelines provide specific recommendations on movement behaviors for children and adolescents. The objective of this study was to verify the adequacy of children and adolescents to the guidelines for moderate to vigorous physical activity, recreational screen time, and sleep duration, and the overall adequacy to the guidelines, before and during the COVID-19 pandemic. A cross-sectional study was conducted with parents or guardians of children or adolescents from different regions of Brazil using a digital interview form including sociodemographic characteristics of families, moderate to vigorous physical activity, recreational screen time, and sleep duration before and during the pandemic. Statistically significant variation was observed in both groups in relation to moderate to vigorous physical activity and recreational screen time between the two periods evaluated. Overall adequacy to the guidelines before the pandemic was 19.28% for children from Group 1 (0–5 years old) and 39.50% for those from Group 2 (6 to 17 years old). During the pandemic, it corresponded to 3.58% in Group 1 and 4.94% in Group 2 (*p*-value between periods ≤0.001). This study showed the significant impact of pandemic restrictions on reducing overall compliance and physical activity, and increasing screen time among Brazilian children and adolescents.

## 1. Introduction

The 24-Hour Movement Guidelines were published by the Canadian Society for Exercise Physiology in 2016 [1]. Subsequently, in 2019, the World Health Organization (WHO) [2] and the Brazilian Society of Pediatrics (SBP) [3] also published their recommendations. The 24-Hour Movement Guidelines aim to offer specific recommendations on the practice of moderate to vigorous physical activity (MVPA), recreational screen time (television, mobile phone, computer, laptop, notebook, or tablet), and adequate sleep duration for children and adolescents according to age group [4]. Despite the recognized importance for health, there are still few studies evaluating adherence to these movement guidelines in low- and middle-income countries [5,6,7,8,9,10].

The 24-Hour Movement Guidelines build on available evidence regarding the importance of all movement behaviors in promoting population health rather than movement behaviors alone [11,12]. The concept that “the whole day matters” is strongly used, showing positive associations with motor development and physical conditioning for children aged 0–4 years, and adiposity reduction and improvement in cardiometabolic conditioning for children aged 5–17 years [13,14].

Since the outbreak of severe acute respiratory syndrome Coronavirus 2 (SARS-CoV-2), the routines of Brazilian children and adolescents have been directly impacted by measures to prevent contagion and combat the spread of the virus [5]. Social distancing and the suspension of classroom classes in government and private schools were encouraged [15], with some establishments offering distance learning. In Brazil, however, as observed in the health sector, the health and educational crises were confronted in a heterogeneous way among different governmental contexts, with inevitable consequences for children and adolescents [16,17]. It is important to note that previous studies have shown that staying at home negatively influences the daily lives of school-aged children and adolescents, since this group tends to be less active on weekends and during holidays and exhibits greater use of screens and electronic devices [18,19,20].

Thus, a sample of Brazilian children and adolescents was evaluated for MVPA practice, recreational screen time, and sleep duration, as well as the overall adequacy to all three components. The objective of this study was to verify the adequacy to the recommendations of the 24-Hour Movement Guidelines at two time points: before and during the COVID-19 pandemic.

## 2. Materials and Methods

### 2.1. Study Design, Configuration, and Participants

This was a cross-sectional study conducted between September 2020 and January 2021. A convenience sample composed of parents or guardians of children and adolescents aged up to 18 years old who agreed to complete a questionnaire electronically sent via social networks was recruited from different geographic regions of Brazil. The non-probabilistic sampling technique (in which the participants are volunteers) was based on a “snowball” data collection process [21]. Snowball sampling is a convenience sampling method that is applied when it is difficult to access subjects with the target characteristics. The population was chosen from the networks of the researchers involved, who were encouraged to forward the electronic form for data collection to their own contacts and so on. Sampling continued until data saturation.

### 2.2. Ethical Considerations

This study was approved by the Ethics Committee for Research with Human Beings of the Fernandes Figueira National Institute of Women, Children, and Adolescents (approval number 4.277.985), and was based on The Checklist for Reporting Results of Internet E-Surveys (CHERRIES) [22] and The Strengthening the Reporting of Observational Studies in Epidemiology (STROBE) guidelines [23]. All participants were informed of their rights and electronically signed the free and informed consent form.

### 2.3. Instrument and Data Collection

The electronic form for data collection was developed on a digital platform (Google Forms) and consisted of 46 variables that were used to collect information on behaviors before and after March 2020, when the COVID-19 pandemic was declared in Brazil. The electronic questionnaire was sent between September and December 2020 to the parents or guardians of children aged 0 to 5 years (Group 1) and between October 2020 and January 2021 to the parents or guardians of children or adolescents between 6 and 17 years old (Group 2). The form contained closed questions and the filling in of the questionnaire by the participants lasted about 20 min.

Sociodemographic variables of the parents and guardians of children and adolescents were collected as follows: region of the country (south; southeast; midwest; north and northeast); age; sex (male; female); ethnicity (white; brown/black); higher education (no; yes); participation in distance learning (no; yes); employment status at the date of completion of the form and variation from March 2020 (employed/unemployed); remote work (no; yes); current average family income (above 10 times the minimum wage; between 4 and 10 times the minimum wage; below 4 times the minimum wage, considering that the Brazilian minimum wage in 2021 was BRL 1100.00); loss of family income from March 2020 (no; yes).

The mean MVPA time, recreational screen time, sleep duration, and the three parameters combined were categorized as adequate or not adequate before and during the COVID-19 pandemic according to the 24-Hour Movement Guidelines. The question about average MVPA time provided five answer options: less than half an hour a day; half an hour a day; 1 h a day; 1.5 h a day; more than 1.5 h a day. The average daily screen time (television, mobile phone, computer, laptop, notebook, or tablet) was assessed using seven options: none; less than 1 h; 1 h; 2 h; 3–4 h; 5–7 h; more than 8 h). Average sleep duration (including naps) was assessed using five options: 15 or more hours a day; 13 to 14 h a day; 11 to 12 h a day; 8 to 10 h a day; less than 8 h a day.

### 2.4. Outcomes

The main outcomes were adequacy to the recommendations of the 24-Hour Movement Guidelines regarding MVPA, recreational screen time according to the type of device (television, mobile phone, computer/tablet), sleep duration, and overall adequacy to these three components [1,4]. The following are the recommendations for each component:MVPA: For infants who are still unable to walk: 30 min throughout the day in the prone position (belly down) or playing sitting away from the screens; for children aged between 1 and 2 years: at least 180 min of MVPA throughout the day; from the age of 3: 60 min of MVPA.Recreational screen time: For children under 2 years: exposure to digital screens is not suitable; between 2 and 4 years: up to one hour a day, with no exposure being better; 5 years or older: not exceeding two hours a day of exposure to digital screens.Sleep duration (including naps): Between 0 and 3 months of age: 14–17 h/day; 4–11 months of age: 12–16 h/day; 1–2 years: 11–14 h/day; 3–5 years: 10–13 h/day; 6–13 years: 9–11 h/day; 14–17 years: 8–10 h/day.Overall adequacy: status of simultaneous adequacy to MVPA, recreational screen time, and sleep duration guidelines for each participant.

### 2.5. Data Analysis

Data were stored in an electronic spreadsheet created automatically by Google Forms, which was exported in Excel format for further analysis using the JASP 0.16.1 statistical package. The categorical variables were described by their respective frequencies of occurrence (absolute and relative) and numerical variables by the means and standard deviations. The chi-square and McNemar’s tests were used to evaluate differences in MVPA adequacy, recreational screen time, sleep duration, and overall adequacy before and during the COVID-19 pandemic.

## 3. Results

From the 525 parents or guardians who participated in this study, answers were obtained regarding 363 children aged 0 to 5 years (Group 1) and 162 children or adolescents aged 6 to 17 years old (Group 2). The mean age of parents or guardians was 37.71 years for Group 1 and 44.21 for Group 2. The remaining sociodemographic characteristics of participants are shown in Table 1.

Table 2 presents the results related to MVPA, recreational screen time, and sleep duration provided by parents or guardians regarding the periods before and during the pandemic.

### 3.1. Moderate to Vigorous Physical Activity

There was a statistically significant reduction (*p*-value < 0.001) of MVPA in both groups observed when comparing the periods before and during the pandemic (Table 2).

### 3.2. Recreational Screen Time

The adequacy to guidelines for recreational screen time was collected according to the type of device (television, mobile phone, computer/tablet). During the pandemic, both groups evaluated showed a statistically significant reduction in the adequacy to recommended screen use time, regardless of the device used (*p*-value < 0.001). Group 1 started using TVs more than any other screen type (Table 2).

In addition to the recreational use of electronic devices during the pandemic, approximately 50% of the children in Group 1 and 92% of the children in Group 2 participated in distance learning. Computers were the most used device for these activities in both groups, with use time exceeding two hours in 11% of Group 1 and 54% of Group 2 (Table 3).

### 3.3. Duration of Sleep

Regarding the adequacy to recommendations on sleep duration, there was no statistically significant variation before and during the pandemic (*p*-value = 0.09) in either group. However, about 50% of the children in Group 1 were adherent to the guidelines before the pandemic, with a small reduction of this percentage (47.38%) during the pandemic period. Children in Group 2 presented 82.72% adequacy before and 86.42% during the pandemic (Table 2).

### 3.4. Overall Adequacy to the 24-Hour Movement Guidelines

There was a statistically significant reduction in the overall adequacy to the 24-Hour Movement Guidelines (MVPA, recreational screen time, and sleep duration) in both groups (Table 2). There was a reduction in overall adequacy in relation to the period evaluated of approximately 5 times in Group 1 and approximately 8 times in Group 2. Table 2 shows the percentages of overall adequacy in each group before and during the pandemic. The percentage of overall adequacy among the entire sample (Groups 1 and 2) was 25.52% before the pandemic and 4% during the pandemic (Table 2).

## 4. Discussion

Based on the information provided by parents and guardians, this study supports the hypothesis that behavioral changes resulting from the COVID-19 pandemic significantly interfered in the overall adequacy to the recommendations of the 24-Hour Movement Guidelines. Objectively, there was a negative impact in relation to physical activity, recreational screen use time, and the overall adequacy to the three components. The inadequacy of the above-mentioned guidelines has been considered to be a risk factor for overweight and for changes in children’s neurocognitive functioning [2].

Regarding the adequacy of individuals to recommended levels of physical activity before and during the pandemic, a reduction was found. Similarly, other studies have reported a decrease in the proportion of children and adolescents who complied with the MVPA guidelines [24,25,26,27,28]. Physical activity reduction is associated with numerous negative health outcomes: hypercholesterolemia, depression, decreased bone density, hypertension, obesity, and metabolic syndrome [29]. Higher levels of MVPA in children are associated with lower adiposity, lower cardiometabolic risk factors, and better cognitive function [30,31].

Regarding screen time, considering the different types of screens, the use of TV presented the greatest decrease in adequacy. These results are aligned with the evidence showing that sedentary behavior increased worldwide during the pandemic [5,9,24,25,32,33,34,35,36,37,38,39,40,41]. In addition, a pattern can be observed in the literature: the increase in screen time was higher for children in countries with stricter social isolation measures, such as Spain, Brazil, and Turkey [5,42,43,44], while it was comparatively lower in countries with mild restrictions, such as Germany, Australia, and Slovenia [28,45,46]. However, regardless of the pandemic, a systematic review pointed out that in children under 2 years of age, the adequacy to screen time guidelines (0 h/d) was 24.7% (95% CI, 19.0–31.5%), and that for children aged 2 to 5 years (1 h/d) was 35.6% (95% CI, 30.6–40.9%) [47].

Regarding sleep time, no statistically significant reduction was observed between the periods evaluated, but the present study produced an unexpected finding: the adequacy to the recommendations was already extremely low before the pandemic, especially in children under 6 years of age. The systematic review by Kharel et al. (2022) evaluated only studies that were conducted during the pandemic and showed variable results regarding sleep duration: increased daily sleep duration among children and adolescents in Chile, Spain, Singapore, and Italy, and preschoolers in China; decreased sleep in the UK and Poland; and no significant difference in sleep duration among children and adolescents before and during the pandemic in other studies in Spain, Australia, USA, Portugal, and China [48]. It is important to highlight that sleep is essential for children’s health and well-being. Inadequate sleep has been associated with worse cognitive outcomes [49] including behavioral changes [50], attention deficit [51], anxiety/depression [52], hyperactivity [53,54], and cardiometabolic problems [55] in children beyond obesity [56,57,58].

Overall adequacy to the recommendations of the 24-Hour Movement Guidelines was already challenging in several countries before the pandemic [6,7,41,59,60,61]. Tapia-Serrano et al. (2022) [62] carried out a meta-analysis that included 63 studies carried out before the COVID-19 pandemic with a sample of 387,437 individuals aged 3 to 18 years from 23 countries. The authors observed that only 7.12% (IC95%: 6.45–7.78) of the individuals conformed to the three components of the 24-Hour Movement Guidelines. On the other hand, the present study observed that, before the pandemic, the overall adequacy of this sample was 25.52%. The data presented by Tapia-Serrano et al. (2022) [62] regarding adequacy during the pre-pandemic period were worse compared to those of the present study, probably due to the high heterogeneity of the included studies.

Other studies have also found a significant reduction in the proportion of children and adolescents of different nationalities who conformed to all of the 24-Hour Movement Guidelines during the restrictions imposed in response to COVID-19 [25,27,30,34,36,39,41,63,64]. These data corroborate those of the present study since 19.28% of the children in Group 1 were considered adherent before the pandemic; however, this percentage reduced to 3.58% during the pandemic. In Group 2, the percentage of overall adequacy was 39.5%, with a reduction to 4.94% during the period of social isolation.

It is important to point out that the recommendations of distance learning and social isolation made in response to COVID-19 affected the participation of children and adolescents in physical education, sports, and other forms of MVPA related to their school or community activities. However, it should be noted that in the adequacy analysis of the present study, the distance learning hours adopted by several schools were not included, because it was understood that there was no viable alternative for formal education. If these hours were included, the percentage of adequacy in the present study would have been even lower.

Online surveys are a promising tool for health research; this survey allowed the participation of individuals from all over the country, especially at a time when it was not possible to conduct face-to-face research due to the social distancing measures in force in Brazil at the time of data collection.

Emerging evidence shows that overall compliance with the 24-Hour Movement Guidelines can lead to health benefits. Considering that the information available on the subject among children and adolescents from low- and middle-income countries is scarce, the strength of this study is in the identification of the overall adequacy to the recommendations of the 24-Hour Movement Guidelines among a Brazilian sample. However, the limitations of this study should be recognized. First, online surveys may be subject to bias resulting from the following issues: (1) the non-representative nature of the internet population; (2) self-selection of participants (voluntary effect), since people who do not have access to the Internet have a null probability of selection and, as the participants were volunteers, the selection probabilities and the non-response rate could not be estimated; and (3) online surveys allow the collection of valuable information during critical periods, but the veracity of the responses in this study may be subject to recall bias, since information on habits prior to the pandemic was requested.

A further limitation is that one in four people in Brazil does not have access to the internet. In total numbers, this represents about 46 million Brazilians (25.3%). In rural areas, the rate of people without access is even higher than in cities, reaching 53.5%. In urban areas it is 20.6% [65]. In addition, we understand that the studied population is representative of the Brazilian middle class, with access to private health and education services, and with high education and income levels. Theoretically, this is a more informed and less vulnerable population. Even so, the study is relevant since a low adequacy was identified even in this sample, prompting consideration the damage to the most vulnerable. Finally, generalization of the results must be carried out with caution, given that most of the participants were from the southeastern region, one of the most developed regions of Brazil.

## 5. Conclusions

The overall adequacy to the recommendations of the 24-Hour Movement Guidelines regarding MVPA, recreational screen time, and sleep duration was not satisfactory before the COVID-19 pandemic, with worse results during the pandemic. Social isolation measures contributed to the reduction of physical activity and to the increase of screen time among Brazilian children and adolescents. Therefore, we recommend education regarding the importance of physical activity, reducing the use of screens, and adequate sleep time according to the age group. The monitoring of actions aimed at children, adolescents, parents, schools, and health professionals should be encouraged to achieve healthier lifestyles and the improvement of habits related to health and school performance beyond the pandemic scenario.

## Figures and Tables

**Table 1 ijerph-20-05737-t001:** Sociodemographic variables of children in Groups 1 and 2, and of their guardians, between 2020 and 2021.

Variables	Group 1 ^1^ (*n* = 363)		Group 2 ^2^ (*n* = 162)	
	Children ^1^	Responsible	Children/Adolescents ^2^	Responsible
	Mean (*SD*)	Mean (*SD*)	Mean (*SD*)	Mean (*SD*)
Age (years)	3.34 (1.55)	37.71 (6.35)	11.27 (3.44)	44.21 (8.19)
	*n* (%)	*n* (%)	*n* (%)	*n* (%)
Sex				
Male	200 (55.10)	26 (7.16)	78 (48.15)	23 (14.20)
Female	163 (44.90)	337 (92.84)	84 (51.85)	139 (85.80)
Ethnicity				
White	-	272 (74.93)	-	125 (77.16)
Black or brown	-	91 (25.07)	-	37 (22.84)
Region of the country				
Southeast	-	309 (85.12)	-	153 (94.44)
Others	-	54 (14.88)	-	9 (5.56)
Higher education				
No	-	41 (11.30)	-	18 (11.11)
Yes	-	322 (88.70)	-	144 (88.48)
Distance learning				
No	182 (50.14)	-	12 (7.41)	-
Yes	181 (49.86)	-	150 (92.59)	-
Employment situation				
Unemployed	-	37 (10.19)	-	15 (9.26)
Employed	-	326 (89.81)	-	147 (90.74)
Change of working regime				
No	-	92 (25.34)	-	38 (23.46)
Yes	-	271 (74.66)	-	124 (76.54)
Remote work				
No	-	112 (30.85)	-	40 (24.69)
Yes	-	228 (62.81)	-	112 (69.14)
Missing	-	23 (6.34)	-	10 (6.17)
Family income ^3^				
<4 m.w.	-	55 (15.15)	-	25 (15.43)
4 and 9 m.w.	-	129 (35.54)	-	54 (33.33)
≥10 m.w.	-	179 (49.31)	-	83 (51.24)
Loss of family income				
No	-	172 (47.39)	-	92 (56.79)
Yes	-	191 (52.62)	-	70 (43.21)

^1^ Children from 0 to 5 years old; ^2^ children and adolescents aged 6 to 17 years; ^3^ Brazilian minimum wage in 2021: BRL 1100.00; *n*: number; *SD*: standard deviation; m.w.: minimum wage.

**Table 2 ijerph-20-05737-t002:** Adequacy of Groups 1 and 2 to the 24-Hour Movement Guidelines (moderate to vigorous physical activity, recreational screen time, and sleep duration) before and during the pandemic.

Group		Group 1 ^1^ (*n* = 363)			Group 2 ^2^ (*n* = 162)		
		Before the Pandemic	During the Pandemic	*p*-Value	Before the Pandemic	During the Pandemic	*p*-Value
Variables		*n* (%)	*n* (%)		*n* (%)	*n* (%)	
MVPA ^3^		223 (61.43)	140 (38.57)	<0.001	118 (72.84)	40 (24.69)	<0.001
Screen time	Television	244 (67.22)	99 (27.27)	<0.001	123 (75.93)	60 (37.04)	<0.001
	Cell phone	319 (87.88)	262 (72.18)	<0.001	117 (72.22)	69 (42.60)	<0.001
	Tablet, computer or notebook	339 (93.39)	306 (84.30)	<0.001	141 (87.03)	98 (60.49)	<0.001
Duration of sleep		185 (50.96)	172 (47.38)	0.09 ^4^	134 (82.72)	140 (86.42)	0.09 ^4^
Overall adequacy		70 (19.28)	13 (3.58)	<0.001	64 (39.50)	8 (4.94)	<0.001

^1^ Children from 0 to 5 years old; ^2^ children and adolescents aged 6 to 17 years; ^3^ moderate to vigorous physical activity; ^4^ McNemar’s signed-rank test. *n*: number.

**Table 3 ijerph-20-05737-t003:** Absolute and relative frequencies among Groups 1 and 2 of electronic device use for school activities during the pandemic, distributed by the hours of daily use.

Group	Group 1 ^1^ (*n* = 181)				Group 2 ^2^ (*n* = 150)			
	Device *n* (%)				Device *n* (%)			
T.	TV	Cell Phone	Tablet	Computer	TV	Cell Phone	Tablet	Computer
0 h	160 (88.4)	107 (59.1)	148 (81.8)	63 (34.8)	145 (96.6)	75 (50.0)	127 (84.7)	20 (13.3)
<1 h	15 (8.3)	52 (28.7)	19 (10.5)	55 (30.4)	1 (0.7)	18 (12.00)	2 (1.3)	7 (4.7)
1–2 h	4 (2.2)	15 (8.3)	10 (5.5)	43 (23.7)	2 (1.3)	36 (24.00)	9 (6.0)	42 (28.0)
3–4 h	2 (1.1)	6 (3.4)	3 (1.7)	11 (6.1)	1 (0.7)	11 (7.3)	7 (4.7)	42 (28.0)
>5 h	0 (0.0)	1 (0.5)	1 (0.5)	9 (5.0)	1 (0.7)	10 (6.7)	5 (3.3)	39 (26.0)

^1^ Children from 0 to 5 years old; ^2^ children and adolescents from 6 to 17 years; T. = time of use; *n*: number.

## Data Availability

Not applicable.

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
