# Peer review of "Survey of the Adequacy of Brazilian Children and Adolescents to the 24-Hour Movement Guidelines before and during the COVID-19 Pandemic"

_ijerph, 2023, doi:10.3390/ijerph20095737_

Round 1
Reviewer 1 Report
Good article.
Please just clarify how adequacy was determined for each aspect and overall adequacy?

Author Response
REVIEWER 1
Dear reviewer, we thank you in advance for your valuable time and suggestions.
Replies below:
line 68- 2.2 Ethical Considerations - number of approval
"the study was approved by the Ethics Committee in Research with Human Beings of the National Institute of Women, Children and Adolescents Fernandes Figueira (CEP-IFF/Fiocruz) (approval number 4.277.985)."
line 77- 2.3 Instrument and data collection
Question: Please clarify- were only 1 form/questionnaire completed for before and after COVID-19 for the 2 different groups. Meaning that the forms were sent on different times to the 2 different groups. Or a before COVID-19 and after form/questionnaire (1 questionnaire) were completed by the 2 different groups?
One single form was sent on different times to the 2 different groups.
"The electronic form for data collection was developed on a digital platform (Google Forms) consisting of 46 variables that were used to collect information on facts that occurred before and after March 2020, when the COVID-19 pandemic was declared in Brazil. The electronic questionnaire was sent between September and December 2020 to parents or guardians of children aged 0 to 5 years (Group 1) and between October 2020 and January 2021 to parents or guardians of children or adolescents between 6 and 18 years old (Group 2). The form contained closed questions and the filling in of the questionnaire by the participants lasted about 20 minutes"
line 123 - 2.5 Data analysis
Question: How was overall adequacy determined? How was the rating done for adequacy regarding the 3 aspects? Please elaborate.
Adequacy was defined as compliance to each separate component (MVPA, recreational screen time and sleep duration) of the 24-h Movement Guideline for each age group.
For overall adequacy: compliance of each participant to all 3 components according to age group.
3.2. Recreational screen time
Question: Please clarify - how this can be said. The % in Table 2 for the use of computers is less in both the groups?
"Computers were the most used device for these activities in both groups, with use time exceeding two hours in 11% of Group 1 and 54% of Group 2
Table 2 refers to adequacy of Groups 1 and 2 to the 24-hour Movement Guidelines (moderate to vigorous physical activity, recreational screen time and sleep duration) before and during the pandemic. Hence, adequacy decreased from 93.3% to 84.3% in group 1 and from 87% to 60.49% in group 2. Therefore, the use of tablet, computer or notebooks increased. However, it sounds confusing, we agree.
For your reference, the table below was not included in the manuscript and refers to the use of different electronic devices. It can be included if appropriate.
Table 3. Absolute and relative frequencies of groups 1 and 2 who used electronic devices for school activities during the pandemic, distributed by the hours of daily use.
|
Group |
Group 11 (n=181) |
Group 22 (n=150) |
||||||
|
|
Device n(%) |
Device n(%) |
||||||
|
T. |
TV |
Cell phone |
Tablet |
Computer |
TV |
Cell phone |
Tablet |
Computer |
|
0h |
160 (88.4) |
107 (59.1) |
148 (81.8) |
63 (34.8) |
145 (96.6) |
75 (50.0) |
127 (84.7) |
20 (13.3) |
|
<1h |
15 (8.3) |
52 (28.7) |
19 (10.5) |
55 (30.4) |
1 (0.7) |
18 (12.00) |
2 (1.3) |
7 (4.7) |
|
1-2h |
4 (2.2) |
15 (8.3) |
10 (5.5) |
43 (23.7) |
2 (1.3) |
36 (24.00) |
9 (6.0) |
42 (28.0) |
|
3-4h |
2 (1.1) |
6 (3.4) |
3 (1.7) |
11 (6.1) |
1 (0.7) |
11 (7.3) |
7 (4.7) |
42 (28.0) |
|
>5h |
0 (0.0) |
1 (0.5) |
1 (0.5) |
9 (5.0) |
1 (0.7) |
10 (6.7) |
5 (3.3) |
39 (26.0) |
1 Children from 0 to 5 years old
2 Children and adolescents from 6 to 18 years incomplete
- = time of use
As for the spelling adjustments of lines 173, 219 and 225, these were corrected as recommended.
Author Response
Reviewer 2
Dear reviewer, we thank you for your valuable time and suggestions for our manuscript.
Replies below:
- Please elaborate regarding the method in which the population was chosen
and approached to complete the online survey.
We used the non-probabilistic sampling technique called “snowball sampling”. The population was chosen from the network of the researchers involved, who were encouraged to forward the electronic form for data collection to their contacts and so on. Social media was also used to disseminate the questionnaire. Participants were also asked to share the interview.
Snowball sampling is a convenience sampling method that is applied when it is difficult to access subjects with the target characteristics. This form of approach was chosen, even considering all the limitations, due to the sanitary scenario at the time, with circulation restrictions and the impossibility of field studies. Sampling took place until data saturation.
- What is the response rate (how many parents were offered to answer and among them, how many actually responded)?
As the study predicted self-selection of participants (voluntary effect), the response rate could not be calculated. As we used a chain dissemination method, especially via WhatsApp and Instagram, and not a fixed list of emails, phone numbers or number of hits to a website, there was no calculation because we did not have a defined sample group. However, we do have data on when and how quickly forms were filled out.
- Please provide any data regarding the medical status of the children represented in this study (for example BMI, obesity, and other general medical conditions). Is there any correlation to the findings in this study?
Although your question is very important, the study did not have the objective of clinical evaluation of the children. Hence, we do not have data on BMI, obesity, and other general medical conditions. However we can inform, according to demographic data collected, these were Brazilian middle-class children, with access to private health and education systems.
- Any data regarding the health status of the parents?
The aim of the study was not to collect clinical data from parents, but we can inform you that they are also people from the Brazilian middle class, with access to a private health and education system.
- The limitations are mentioned in the discussion, but it should be emphasized that the population is non representative as compared to the general population.
After your comment, we will include the following excerpt:
“Still as a limitation, we point out that one in four people in Brazil does not have access to the internet. In total numbers, this represents about 46 million Brazilians (25.3%). In rural areas, the rate of people without access is even higher than in cities, reaching 53.5%. In urban areas it is 20.6% (IBGE, 2018). In addition, we understand that the studied population is representative of the Brazilian middle class, with access to private health and education services, with high education and income. Theoretically, this is a more informed and less vulnerable population. Even so, the study is relevant since in this sample a low adequacy has already been identified, making us think about the damage to the most vulnerable. Finally, the generalization of the results must be done with caution, given that most of the participants were from the Southeast region, one of the most developed regions of Brazil.”
BRAZIL, Brazilian Institute of Geography and Statistics (IBGE). Continuous National Household Sample Survey - Access to the Internet and television and possession of a cell phone for personal use 2018 [Internet]. 2020 [cited December 12, 2021]. Available at: https://biblioteca.ibge.gov.br/visualizacao/livros/liv101705_informativo.pdf
Reviewer 3 Report
The manuscript presents a relevant theme for publication in IJERPH, which may be accepted with some minor revisions.
- In my opinion, the introduction provides information to establish the research questions raised in the manuscript; however, I think that the state of the art about the study problem could be more developed; In fact, the discussion cites several studies that could be developed in the literature review.
- The methodology provides enough details; however, the sample description could be more objective.
- The results section is sufficiently clear and precise.
- the discussion/conclusions of results based on previous literature.
After carefully reading your manuscript, I point out some aspects that must be improved and corrected:
1. The authors state, “... The categorical variables were described by their respective frequency of occurrence (absolute and relative) and numerical variables by median and interquartile range.” However, by analyzing the data, the authors only used the age variable's median and interquartile range. So why didn't the authors use the mean and the standard deviation?
2. All statistical symbols must be in italics (n, p, …. ).
3. Some formatting aspects should be corrected (spelling, punctuation). Please, correct what is pointed out in the body of the manuscript.

Author Response
Reviewer 3
Dear reviewer, thank you for valuable time and contributions to the manuscript.
Modifications suggested by Reviewer 3 below (highlighted in yellow in the manuscript) :
- Abstract: Overall adequacy to the guidelines before the pandemic were 19.28% for children from Group 1 (0-5 years old) and 39.50% for Group 2 (6 to 18 years old). During the pandemic, it corresponded to 3.58% in Group 1 and 4.94% in Group 2 (p-value between periods = <0.001).
- Keywords: Guideline Adherence; Exercise; Screen Time; Sleep; Infant; Preschool; Child; Adolescent.
Authors: As for the keyword “exercise”, this was chosen because it is the MESH term for Physical activity, our analyzed variable, as well as Screen Time and Sleep. Although movement is associated with behaviors, as a descriptor it ends up becoming more abstract.
- Since the outbreak of severe acute respiratory syndrome Coronavirus 2 (SARS-CoV-2), the routine of Brazilian children and adolescents have been directly impacted by measures to prevent contagion and combat the spread of the virus [5].
- Thus, a sample of Brazilian children and adolescents was evaluated for MVPA practice, recreational screen time and sleep duration, as well as the overall adequacy of everyone to these three components
- The mean time of MVPA, recreational screen, sleep, and the three parameters altogether, before and during the COVID-19 pandemic were categorized as adequate and not adequate according to the 24-Hour Movement Guidelines.
- 1. MVPA: For infants who are still unable to walk, 30 minutes throughout the day in the prone position (belly down) or playing sitting away from the screens; for children aged between one and two years, at least 180 minutes of MVPA throughout the day; from the age of three, 60 minutes of MVPA.
- The remaining sociodemographic characteristics of participants are shown in Table 1.
- 3.2. Recreational screen time
- The adequacy of recreational screen time was made available by the type of device (television, mobile phone, computer/tablet).
- Regarding screen time, considering the different types of screens, the use of TV was the one that presented the greatest decrease in adequacy.
- The overall adequacy of the recommendations of the 24h…
- Considering that the information available on the subject in children and adolescents from low-and middle-income countries is scarce, the strength of this study is to identify the overall adequacy to the recommendations of the 24-hour Movement Guidelines in a Brazilian sample.
- Finally, the generalization of the results must be made with caution, given that most of the participants were from the Southeast region, one of the most developed regions of Brazil.
Reviewer 3 comments:
I think that the state of the art about the study problem could be more developed; In fact, the discussion cites several studies that could be developed in the literature review.
As a way of presenting the state of the art, as suggested, we will include the following paragraph in the introduction:
“The 24-Hour Movement Guidelines build on available evidence regarding the importance of all movement behaviors in promoting population health rather than movement behaviors alone (Chaput et al, 2014; Saunders et al, 2016). The concept that “the whole day matters” is strongly used, showing positive associations with motor development and physical conditioning for children aged 0-4 years, adiposity reduction and improvement in cardiometabolic conditioning for children aged 5–17 years (Grgic et al, 2018, Kuzik et al, 2017) “
- The methodology provides enough details; however, the sample description could be more objective. Done, thank you.
- the discussion/conclusions of results based on previous literature. Done, thank you.
After carefully reading your manuscript, I point out some aspects that must be improved and corrected:
- The authors state, “... The categorical variables were described by their respective frequency of occurrence (absolute and relative) and numerical variables by median and interquartile range.” However, by analyzing the data, the authors only used the age variable's median and interquartile range. So why didn't the authors use the mean and the standard deviation? Done, thank you.
- All statistical symbols must be in italics (n, p, …. ). Done, thank you.
- Some formatting aspects should be corrected (spelling, punctuation). Please, correct what is pointed out in the body of the manuscript. Done, thank you.